# Optimized Feature Extraction for Sample Efficient Deep Reinforcement Learning

**Yuangang Li [1][iD], Tao Guo [2], Qinghua Li [2] and Xinyue Liu [2,\*]**

1   Faculty of Business Information, Shanghai Business School, Shanghai 200235, China; yuangang.li@sbs.edu.cn
2   School of Software, Dalian University of Technology, Dalian 116620, China; guotao@mail.dlut.edu.cn (T.G.)
\*   Correspondence: xyliu@dlut.edu.cn

**Abstract:** In deep reinforcement learning, agent exploration still has certain limitations, while low efficiency exploration further leads to the problem of low sample efficiency. In order to solve the exploration dilemma caused by white noise interference and the separation derailment problem in the environment, we present an innovative approach by introducing an intricately honed feature extraction module to harness the predictive errors, generate intrinsic rewards, and use an ancillary agent training paradigm that effectively solves the above problems and significantly enhances the agent's capacity for comprehensive exploration within environments characterized by sparse reward distribution. The efficacy of the optimized feature extraction module is substantiated through comparative experiments conducted within the arduous exploration problem scenarios often employed in reinforcement learning investigations. Furthermore, a comprehensive performance analysis of our method is executed within the esteemed Atari 2600 experimental setting, yielding noteworthy advancements in performance and showcasing the attainment of superior outcomes in six selected experimental environments.

**Keywords:** deep reinforcement learning; feature optimization; auxiliary agent; sample efficiency; exploration strategy





## 1. Introduction

In recent times, deep reinforcement learning (DRL) has attained remarkable triumphs across a diverse range of domains, including Go [1,2], Atari [3], StarCraft [4,5], and Robot [6]. This unequivocally unveils the profound potential of deep reinforcement learning, which is widely recognized as the most auspicious solution for real-world sequential decision-making predicaments. However, amid their accomplishments in numerous arenas, one pivotal quandary persists: the proclivity of the deep reinforcement learning method to be unduly inefficient with samples [7], necessitating millions of interactions even in ostensibly uncomplicated game scenarios. For instance, though Agent57 [8] stands as the premier deep reinforcement learning algorithm capable of surpassing the average human player across all 57 Atari games, it generally mandates orders of magnitude more interactions than its human counterpart. The crux of the sample inefficiency quandary lies in achieving a delicate equilibrium between exploring and exploiting the active quest for uncharted states and behaviors that hold the promise of yielding elevated rewards and long-term gains [9], juxtaposed against the knowledge acquired thus far to maximize instantaneous returns. The key conundrum revolves around how an agent ought to judiciously navigate the trade-off between venturing into novel actions and selecting the optimal course of action based on its accrued wisdom. A superlative strategy pertaining to exploration can effectively unravel the enigmatic realms of the unknown environment, facilitating the accumulation of informational experience, thereby hastening the agent's learning pace, accelerating convergence, eliciting greater rewards, and amplifying the overall performance of the agent.

However, current approaches suffer from some problems when uncontrollable Gaussian noise permeates the visual domain, whereby only a minute fraction of the pixel space actually comprises pertinent, utilitarian imagery. Consequently, the agent's capacity to accurately assess the present state is compromised, impeding its ability to select the appropriate course of action for exploration. And, regrettably, existing exploration algorithms persist in exploring while the agent progresses toward the target state, thereby causing a deviation from the original trajectory and ultimately thwarting the agent's arrival at the exploration boundary.

The concept of constructing intrinsic rewards predicated upon predictive models was originally proposed in 1991. A profoundly intuitive approach entails leveraging forward dynamic prediction models to generate prediction errors, which can be formally captured as follows:

$$f(s_t, a_t) \rightarrow s_{t+1} \tag{1}$$

where the model takes the present state $s_t$ and the current action $a_t$ as inputs, employing either a linear function or a neural network to approximate the ensuing state $s_{t+1}$ as dictated by the environment. This model epitomizes the agent's aptitude to prognosticate the consequences of its actions. Naturally, prediction models engender prediction errors within certain states:

$$e(s_t, a_t) = |f(s_t, a_t) - s_{t+1}|_2^2 \tag{2}$$

Consequently, these prediction errors, specifically $e(s_t, a_t)$, can be harnessed to furnish the agent with an intrinsic reward for exploration. The magnitude of this reward is inversely correlated with familiarity pertaining to the current state. Higher values of $e(s_t, a_t)$ signify a lesser degree of acquaintance, thereby meriting augmented rewards intended to encourage exploratory endeavors within unfamiliar territories.

For state-finite Markov decision processes, several rudimentary exploration heuristics are available, such as the epsilon-greedy strategy [10] and entropy regularization [11]. Nevertheless, with the amalgamation of deep learning and reinforcement learning techniques in recent years, certain methodologies have employed neural networks to evaluate Markov decision-making strategies. While they have attained remarkable achievements in various domains, they soon encountered the predicament posed by exceedingly prodigious state spaces. The traditional approach of storing pertinent information concerning the Markov decision process via tabulation becomes untenable, owing to the vast number of states involved. In these instances, conventional exploration strategies fail to yield efficacious outcomes, as the agent finds itself ensnared within a limited subset of states. Moreover, reinforcement learning algorithms are meticulously tailored and assessed within simulated environments replete with dense rewards. However, the real-world milieu is characterized by a scarcity of rewards, with the agent exclusively updating its policies upon reaping rewards [12]. Consequently, the agent's exploration strategy is bestowed with the added responsibility of adroitly eschewing perilous states, thereby imposing more exacting demands. This necessitates not only astute discernment of sparse rewards but also a cautious avoidance of hazardous circumstances.

The challenges posed by deep reinforcement learning dilemmas are further compounded as the state–action space burgeons. Consider, for instance, real-world robots equipped with high-dimensional state inputs like images or high-frequency radar signals [13], coupled with intricate operations mandating an extensive range of degrees of freedom. In essence, profound state–action spaces impede the efficacy and robustness of deep reinforcement learning algorithms. In more complex scenarios, the state–action space may exhibit a labyrinthine underlying structure, replete with causal dependencies between states or a prescribed order of access, wherein certain states are accessed with disparate probabilities. Furthermore, unlike the conventionally investigated continuous or uniformly distributed action spaces, actions may encompass a combination of discrete and continuous components. These pragmatic quandaries pose even more formidable challenges to the realm of efficient exploration.

Real-world environments often exhibit a high degree of randomness, wherein unforeseen elements frequently manifest within both their state and action spaces. To illustrate, consider the visual observations of a self-driving car [14,15], which may include extraneous details such as the shifting positions and contours of clouds. In certain exploratory methodologies, white noise is commonly employed to generate states of heightened entropy, infusing the environment with an element of unpredictability [16,17].

Following extensive training, it becomes necessary to diminish the novelty associated with frequently recurring states, along with the exploration rewards assigned to them. However, empirical investigations have uncovered a predicament arising in specific experimental settings, whereby the rapid decay of exploration rewards engenders additional challenges. For instance, consider a maze game environment composed of numerous diminutive chambers, wherein the agent consistently respawns within a singular, distinctive room. Each iteration necessitates the agent's departure from its initial confinement, proceeding to explore the wider expanse. Yet, when the number of iterations surpasses a certain threshold, the exploration reward assigned to the path leading away from the original room diminishes to such an extent that the agent is incapable of exiting its initial confinement.

The Go-Explore algorithm [18] encapsulates the aforementioned quandaries and identifies two fundamental issues plaguing contemporary curiosity-driven exploration approaches: "separation" and "derailment". The concept of separation denotes that while an exploration algorithm can incentivize the agent to traverse uncharted regions of the state space, it fails to motivate the agent to transcend the boundaries established by prior exploration rewards and venture toward novel frontiers for continued exploration. A discernible "separation" exists between the current state and unexplored states. To address this separation predicament, an intuitive solution emerges promoting the agent to return to a previously explored state boundary before embarking on exploration anew.

We mainly aim to solve the above-mentioned white noise and separation derailment problems. By using the optimized feature extraction module and adding auxiliary agents to form a new training paradigm, we focus on improving the model performance of the agent in the case of sparse rewards and obvious environmental changes. Through the improvement of the above method we can make the agent not fall into the exploration dilemma in the difficult exploration environment and still maintain a better performance.

## 2. Related Work

In sparse-reward environments, it is often imperative to incorporate intrinsic reward signals [19] in order to augment the overall reward acquired by the agent, thereby fostering exploration of the environment and eventual attainment of the ultimate objective. The methods for executing exploration strategies can be broadly categorized into three types, contingent upon the manner in which the intrinsic rewards are obtained: (1) exploration based on state counting; (2) exploration based on information enhancement; and (3) exploration based on curiosity rewards.

### 2.1. Exploration Based on State Counting

Exploration strategies rooted in state counting rely on tallying state–action pairs by converting these counts into rewards. The UCB algorithm [20] (Upper Confidence Bounds) stimulates model exploration by selecting the action that maximizes the reward. Contrasted with the direct utilization of environmental rewards, the UCB method amplifies the likelihood of states with fewer visits. The MBIE-EB method [21] (Model-Based Interval Estimation–Exploration Bonus) utilizes tables to tabulate state–behavior pairs and consequently appends supplementary reward signals to instigate exploration of states with scant visitation.

Divergent from the conventional counting-based approach, a pseudo counting technique based on the state density model [22] has been proposed alongside the UCB framework. The pseudo count is assigned via the construction of a density model, subsequently

employed to calculate additional rewards. DQN + SR [23] (Deep Q-Network + successor representation) employs the norm of the successor representation as an intrinsic reward, showcasing superior performance relative to the density model within continuous spaces.

The aforementioned methodologies aid the training process by incorporating auxiliary rewards in the form of state counts alongside the primary environmental rewards. However, when confronted with a white noise environment, deploying Gaussian noise devoid of any informative value for learning as a novel state serves to spur the model towards exploration, inadvertently leading it astray from the intended goal. Our proposed approach effectively mitigates the issue of white noise through the utilization of an optimized feature extraction module, thereby endowing this methodology with enhanced efficacy in challenging exploration environments.

### 2.2. Exploration Based on Information Enhancement

The pursuit of information-enriched exploration propels the agent to embark on its quest by harnessing the intrinsic reward of information gain while diminishing the allure of random regions. Information gain, a reward bestowed upon the reduction of environmental uncertainty, serves as their compass. Their objective is to glean novel insights as they traverse uncharted states, identifying those endowed with greater potential for information gain as the most coveted destinations.

VIME [24] (Variational Information Maximizing Exploration) aspires for each expedition to amass a bounty of environmental knowledge. This method employs variational inference within the Bayesian neural network framework to formalize the process of learning. Hierarchical reinforcement learning [25] partitions the policy into two components, the primary policy and the sub-policy, wherein the former manipulates the latter, which in turn dictates the original action. A subsequent approach proposes acquiring the optimal strategy for the exploration mechanism by solving the alternative Markov decision process [26], thus ensuring the safety of exploration while attaining an improved exploration strategy.

The aforementioned methods fortify agent training by incorporating information gain as a supplementary reward during the model learning phase, thereby mitigating the impact of random states on the agent throughout the training process. Nevertheless, when confronted with issues of reconnection and derailment, such methodologies prove inadequate in liberating the agent from the quagmire of learning stagnation and pushing it towards the frontier of uncharted environments. Our method resolves this quandary by aiding the agent via collaborative training. With the inclusion of auxiliary rewards, the arduous aspects of exploration are significantly diminished, affording the agent the means to engage in superior exploratory endeavors.

### 2.3. Exploration Based on Curiosity Rewards

Curiosity-based reward exploration entails the formulation of an intrinsic reward system predicated on calculating the disparity between the predicted state and the real state, thereby quantifying the prediction error and guiding the agent's exploration of the environment.

The ICM module [27] confers intrinsic rewards to the agent based on curiosity, employing a forward model to forecast forthcoming states and utilizing the disparity between predicted and actual states as an additional intrinsic reward. ECR [28] (Episodic Curiosity through Reachability) proposes an intrinsic reward mechanism grounded in episodic reachability, whereby diverse intrinsic rewards are dispensed by comparing the reachability of the current state with previously encountered states stored in memory.

Our method leverages the contrast between future and predicted states as an intrinsic reward, optimizing feature embeddings to enhance the model's predictive capabilities, while employing an auxiliary agent to circumvent the predicament of agents becoming disoriented within the environment.

*2.4. Random Distillation Network*

In the realm of research pertaining to the exploration algorithm based on prediction errors, certain scholars have discovered that prediction tasks unrelated to the environmental dynamics can still facilitate the agent's expeditions within the said environment. Amongst these methodologies, Random Network Distillation (RND) [29] stands as an exemplary representation. The modus operandi of RND involves incorporating a prediction task independent of the reinforcement learning objective. This entails designing two neural networks equipped with identical structures for the prediction task:

Target network: denoted as $f : S \rightarrow R^k$ wherein the network parameters $\theta$ are randomly initialized and fixed. It receives the current state $s_t$ as input and outputs a predetermined value $f(s_t)$.

Prediction network: denoted as $\hat{f} : S \rightarrow R^k$ wherein the network parameters $\theta$ are randomly initialized. It receives the current state $s_t$ as input and produces a corresponding prediction of the aforementioned fixed value $f(s_t)$, namely $\hat{f}(s_t)$.

The procedure unfolds as follows: predicting the target network $f(s)$ by evaluating the network $\hat{f}(s)$. For any given state $s_t$ in time, the final outcome $\hat{f}(s)$ is derived and utilized as input for the neural network. Subsequently, it undergoes evaluation by the deterministic function $f(s)$ to obtain the result $f(s_t)$. Consequently, the discrepancy between the two outcomes is quantified as the RND exploration reward under state $s_t$, formulated as:

$$r_t^i = \left|\left| \hat{f}(s_t) - f(s_t) \right|\right| \tag{3}$$

The network error also serves as the loss function for the prediction task, enabling the update of the neural network $\hat{f}(s|\hat{\theta})$ throughout said task. Therefore, training the RND model typically consists of two stages, each corresponding to the prediction task and the reinforcement learning task. The model derived from the prediction task is subsequently employed in training the reinforcement learning model, with both stages conducted alternately.

This paper's methodology builds upon the enhancements made to the RND approach. It introduces an optimized feature extractor to generate an intrinsic reward module, referred to as ESSRND (Enhanced Self-Supervised Random Network Distillation), which aids in agent training. Furthermore, it leverages the auxiliary agent training framework (E2S2RND) to further enhance the method's performance.

## 3. Method

We improve upon RND and propose a new agent training scheme to address the common derailment problem in intrinsic reward-based exploration methods. The state feature extraction module is used to model the model observations, and before using the RND to calculate the intrinsic rewards, similar states can be distinguished to improve the accuracy of generating intrinsic rewards based on prediction errors. In the training phase, an auxiliary agent is introduced, and the interaction data with the environment is generated when the main agent falls into a difficult exploration through trajectory playback and random exploration. The following sections introduce the exploration model ESSRND and the agent training framework E2S2RND proposed in this paper from these two improvement directions.

*3.1. ESSRND*

We combine the optimized feature extractor with the RND original network to construct a new intrinsic reward generation module ESSRND. First, the feature representation $\varphi(s_t)$ of the current state is obtained through the trained feature extractor. According to the division of the feature space, in the optimized feature representation, we remove the state features that the agent cannot control and that do not affect the part of the agent. Then $\varphi(s_t)$ replaces $s_t$ as the input of the target network and prediction network in RND. Its prediction error is calculated and the corresponding exploration reward is generated, as shown in Formula (4):

$$r_t^i = \left|\left| \hat{f}(\varphi(s_{t+1})) - f(\varphi(s_{t+1})) \right|\right| \tag{4}$$

The features of the current state indicate that $\varphi(s_t)$ cancels the process of rounding up and down, and uses the LeakeyReLU activation function in the last layer of the embedded network. The modified feature embedding network module structure is shown in Figure 1.

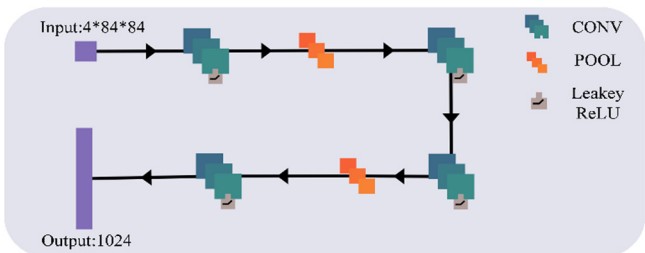

**Figure 1.** Embedded network module structure.

Therefore, the training phase of the prediction task needs to be divided into two parts: (1) Embedding network update: the interaction data between the agent and the environment is obtained by sampling from the Replay-Buffer, and the parameter $\theta_e$ of the feature embedding network is optimized according to the error of the action prediction; (2) Prediction network update: the parameter $\theta_e$ of the embedding network is fixed, and the parameter $\theta_p$ of the prediction network is optimized according to the generated exploration reward error (Formula (4)). The formula of the loss function can be rewritten as:

$$L(s_t, a, s_{t+1}) = NLLLoss(a, a') \tag{5}$$

The agent training phase is the same as in RND, using RND to generate the intrinsic reward function, and sampling the update strategy $\pi$ from the Replay-Buffer:

$$E_{\pi(s_t;\theta_\pi)}\left[ \sum_t \left( r_t + r_t^i \right) \right] \tag{6}$$

By introducing a feature extraction module, the influence of white noise in the original state space is reduced. And the error prediction of auxiliary tasks is performed in the optimized feature space, which can better distinguish similar or different states, and the generated exploration rewards are more meaningful. Thereby this improves the overall exploration ability and performance of the agent. The overall network model of the integrated ESSRND is shown in Figure 2.

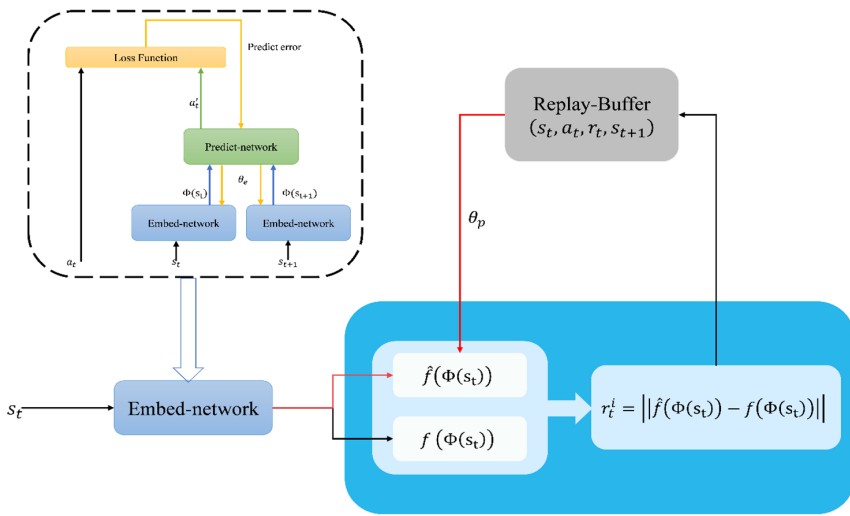

**Figure 2.** Neural network model of ESSRND.

### 3.2. E2S2RND

To address the separation and derailment issues, this paper proposes a novel agent training framework and combines it with ESSRND. The agent is made to generate exploration rewards according to the prediction error in the process of interacting with the environment. When an exploration dilemma occurs, the auxiliary agent is used to interact with the environment, and the main agent is updated according to the interaction fragments of the auxiliary agent to optimize its strategy to obtain better results.

The agent training framework (Figure 3) proposed in this paper can be divided into two stages: In stage one, the original training method is adopted. The agent continuously interacts with the environment, generates exploration rewards based on prediction errors through the ESSRND module, stores all interaction data in the Replay-Buffer, and updates the agent strategy by sampling selected data in the subsequent process. For a state and trajectory with high novelty, it needs to be stored in the Trajectory-Memory according to the rules. Trajectory-Memory is a storage space of size M. In the process of interaction between the agent and the environment, the stored judgment is made according to the following formula:

$$r_t^i > \left(r_k^i\right)_{k=1}^{\{M\}}, s_t \notin S_T \tag{7}$$

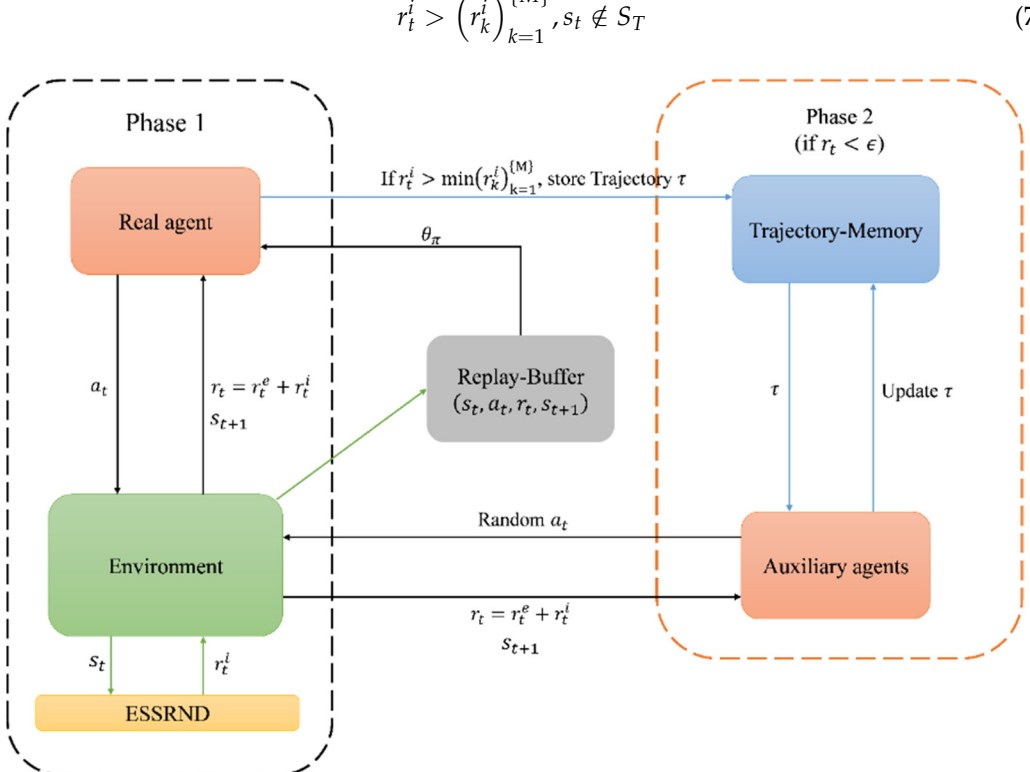

**Figure 3.** The framework of E2S2RND.

Among them, $S_T$ represents the termination state set. If the exploration reward of the current state is greater than the minimum value in the storage space, it is added to the storage space and its action track is recorded.

Phase two starts when the agent receives less reward from the environment than a set threshold $\varepsilon$. In phase two, we use an auxiliary agent for subsequent exploration. First, the corresponding trajectory is taken out from the Trajectory-Memory, and the auxiliary agent returns to the exploration boundary $s_t$ according to the action trajectory in the memory. The auxiliary agent performs random exploration in this state and stores the interaction data in the Replay-Buffer. Here multiple auxiliary agents can be used simultaneously to interact with the environment. After all the auxiliary agents finish exploring, samples are drawn from the Replay-Buffer to train the main agent. After the training is completed, the first stage is restarted, and the two stages are carried out alternately in the subsequent

training. The entire process of the E2S2RND algorithm proposed in this paper is shown in Algorithm 1.

---

**Algorithm 1.** E2S2RND

---

1: Input:
2:     Initialize the parameter $\theta_\pi$ of the policy network, the parameter $\theta_e$ of the embedded network, and predict the network parameter $\theta_p$.
3:     Initialize Trajectory-Memory, size $M_t$; initialize Replay-Buffer, size $M_b$.
4: Iteration:
5:     Pre-trained embedded network model;
6:         **for** each iteration $j$ **do** {
7:         The agent interacts with the environment and collects $(s_t, a_t, r_t, s_{t+1})$ data based on the current strategy $\pi$;
8:         **if** $r_t^i > \left(r_k^i\right)_{k=1}^{\{M\}}$ **then** {
9:             Store trajectory and exploration reward information in Trajectory-Memory;
10:        }
11:        **if** $r_t < \varepsilon$ **then** {
12:            Start the auxiliary agent, select a trajectory from Trajectory-Memory and backtrack;
13:            Perform random exploration from the exploration boundary, and store the interaction record in the Replay-Buffer;
14:            All auxiliary agents are stopped, and the main agent is trained by sampling from the Replay-Buffer;
15:        }
16:        Store the collected samples in the Replay-Buffer;
17:        **if** $j \bmod e_{update} = 0$ **then** {
18:            Sampling from the Replay-Buffer and updating the prediction neural network according to the loss function of Formula (4);
19:            Sampling from the Replay-Buffer and updating the embedded neural network according to the loss function of Formula (5);
20:        }
21:        **if** $j \bmod \pi_{update} = 0$ **then** {
22:            Sampling from the Replay-Buffer to train the main agent;
23:        }
24:    }

---

## 4. Results

We first verified the effectiveness of the ESSRND algorithm in a random noise maze environment. The random noise maze is a reinforcement learning environment constructed based on the pycolab game engine, and it is a typical white noise environment. By testing its exploration ability in the experimental environment and comparing it with other methods, it is verified that the ESSRND method can effectively solve the white noise problem and has better performance for difficult exploration problems. Subsequent experiments were mainly carried out on the Atari 2600, and were carried out for many of the experimental environments that were difficult to explore. The performance of the game was mainly tested in the Montezuma's Revenge environment [30], and related hyperparameter experiments were carried out. The performance of the E2S2RND algorithm was compared and analyzed through the Atari 2600 experimental environment, and it was verified that the E2S2RND algorithm trained with an auxiliary agent can effectively improve the agent's exploration ability and alleviate the impact of "separation" and "derailment" problems on the agent.

### 4.1. ESSRND Effectiveness Experiment

Compared with the original RND, we introduced the state feature extraction module to solve the white noise problem. In order to verify its effectiveness, related experiments were carried out in the random noise maze experimental environment.

Based on the setting of color transformation, the original state space is not large, but the probability of visiting two identical states is extremely low, thus transforming this problem into a difficult exploration problem. The experimental environment is shown in Figure 4.

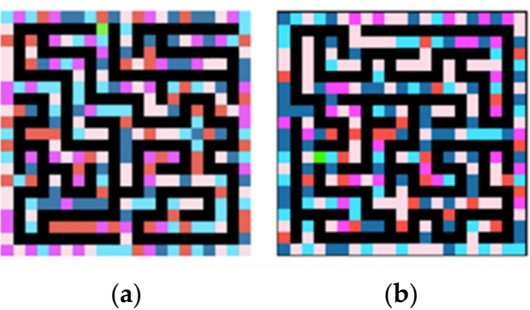

(**a**)          (**b**)

**Figure 4.** The random noise maze is a 21 × 21 grid maze, in which the green square represents the agent, the black represents the passage in the maze, and other colors (including five colors in total) represent the walls. The maze is regenerated every time it is initialized. The agent can choose four actions: left, right, up, and down, and the walls of the maze undergo random color changes after each action. When the agent touches the wall, it directly restarts the next round. (**a**,**b**) represent different moments in the maze.

In the experiment, the ratio of the explored space to the total size of the state space is used as the evaluation index of the agent's exploration ability. The experimental results are shown in Figure 5. Due to the effect of color transformation in the environment, both the randomly embedded features (blue) and the original RND (green) are trapped in white noise, and cannot learn an effective policy. Optimizing feature embedding (orange) can filter out the influence of white noise and increase the exploration utilization rate of the environment as the number of agent interaction steps increases, allowing the gradual learning of useful strategies. Our method can effectively alleviate the white noise problem in the training environment. Compared with random feature embedding and original RND, ESSRND can not only filter out the environment features irrelevant to the agent through optimized feature embedding, but also reach convergence.

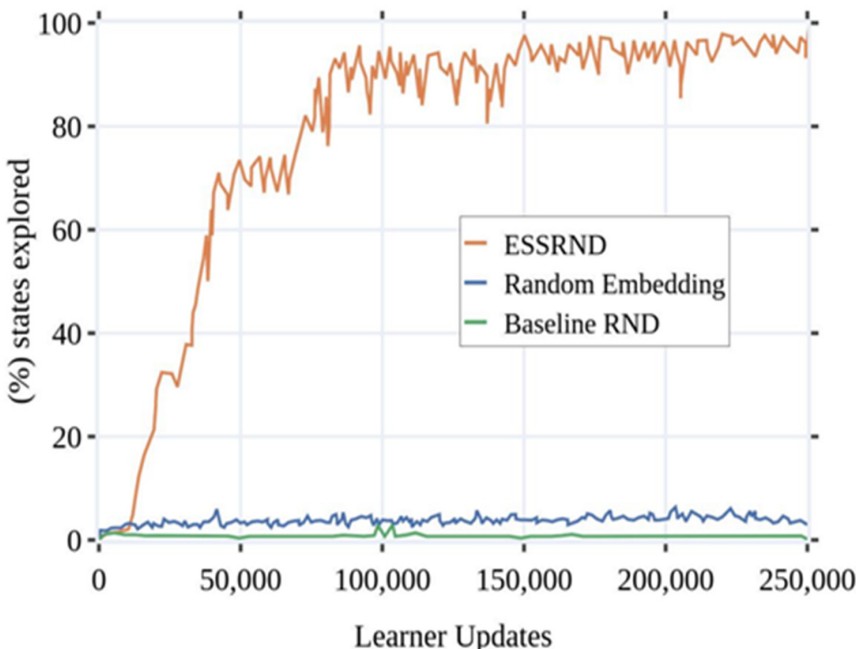

**Figure 5.** Experimental result in Random Disco Maze. In the figure, the original RND (green curve: Baseline RND), random feature embedding (blue curve: Random Embedding) and optimized feature embedding (orange curve: ESSRND) are used for comparison, where the horizontal axis represents the number of agent interaction steps, and the vertical axis represents the environment state exploration utilization.

The background in the maze environment is changing all the time, which greatly affects the ability of the agent to judge the path, so the ability of feature extraction is particularly important. Due to the addition of an optimized feature extraction module in this paper, the ability of the encoder to extract features has been enhanced, and adding the intrinsic rewards described in this paper on the basis of environmental rewards can help the agent to explore better in the maze, and the function convergence is relatively stable and is not affected by environmental changes.

### 4.2. E2S2RND Algorithm Performance Experiment

We conducted comparative experiments in the Montezuma's Revenge experimental environment, mainly comparing the performance of E2S2RND and the original RND method, and the experimental results are shown in Figure 6. As can be seen from the figure, our proposed ESSRND method (green) and the auxiliary agent training framework E2S2RND (orange) have a significant performance improvement compared to the original RND method (red). Moreover, the convergence speed of our method during the training process is also slightly improved compared with the original RND method. E2S2RND has higher rewards than ESSRND in most parameter update steps, and E2S2RND maintains a certain upward trend compared with the peak value of ESSRND. It can be seen that after applying the E2S2RND algorithm training framework, the performance of the agent is still subject to a degree of performance improvement.

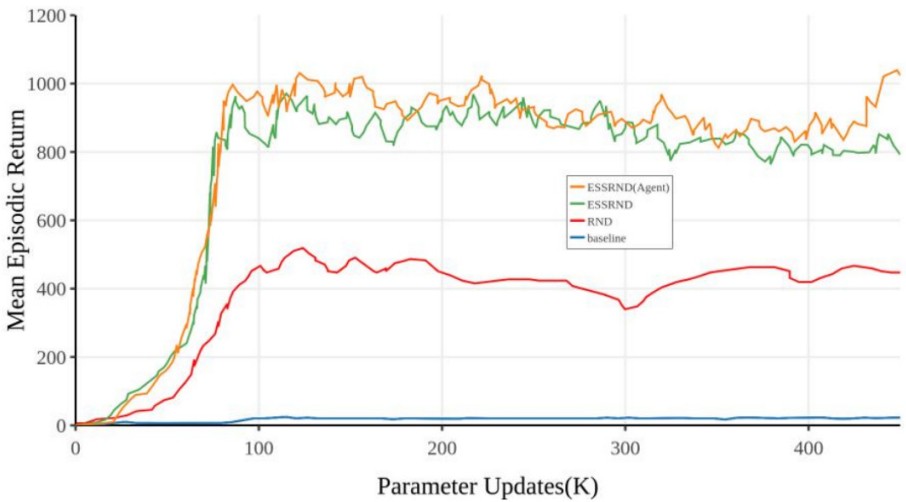

**Figure 6.** Experimental results in Montezuma's Revenge. In the figure, we compare the performance of four methods, including baseline (blue curve), original RND (red curve), ESSRND optimized feature embedding (green curve) and E2S2RND assisted agent algorithm (orange curve). The horizontal axis represents the number of model parameter update steps, and the vertical axis represents the average reward during the agent interaction process.

This result can prove that using the auxiliary agent training framework (E2S2RND) for training on the basis of ESSRND can help the agent understand the derailment problem, jump out of the predicament, make it better for performing exploration in unknown areas, and improve the learning ability of the agent. Using this training framework can also increase the upper learning limit of the agent, so that it can finally achieve better performance.

We also conducted comparative experiments for different methods, and the results are shown in Table 1. The experimental results of the optimized feature embedding ESSRND and the auxiliary agent training framework E2S2RND are all better than the original RND baseline method. It can be concluded that our method has greatly improved the final performance of the agent. After using the auxiliary agent framework for training, the experimental results are better than the ESSRND method without framework training, and achieve the best performance in the Pitfall, PrivateEye, Montezuma, and Venture

environments, so our proposed auxiliary agent training framework has a certain role in difficult exploration environments and can help agents learn more diverse data and then learn better strategies.

**Table 1.** Performance comparison experiments on Atari games. Comparative experiments were carried out on six environments such as Gravitar and Montezuma's Revenge [30]. Among them, the original RND algorithm is used as the benchmark for comparison. Avg.Human represents the average score of humans; ESSRND and E2S2RND are the results of our method. In the experiment, 10 M interactions are performed on each environment.

|  | Pitfall | PrivateEye | Gravitar | Montezuma | Solaris | Venture |
|---|---|---|---|---|---|---|
| RND [29] | −3 | 8666 | 3906 | 8152 | 3282 | 1859 |
| PPO [31] | 0 | 105 | 3426 | 2497 | 3387 | 0 |
| Dynamics | 0 | 33 | 3371 | 400 | 3246 | 1712 |
| R2D2 [32] | −0.19 | 30,345 | 7090 | 2666 | **17,741** | 1958 |
| NGU [33] | 7800 | 65,600 | **14,200** | 8900 | 4400 | 1700 |
| Avg.Human | 6464 | 69,571 | 3351 | 4753 | 12,327 | 1188 |
| ESSRND | 8403 | 58,776 | 8349 | 9237 | 4632 | 1901 |
| E2S2RND | **9421** | **83,224** | 11,274 | **10,436** | 5568 | **1984** |

The R2D2 method [29] introduces Behavior Transfer (BT), a technique that utilizes pretrained policies for exploration. Compared with the R2D2 method, our method is higher than the R2D2 method except for the experimental results on the Solaris environment.

The NGU method [30] uses the same neural network to simultaneously learn multiple directed exploration strategies, with different trade-offs between exploration and exploitation. The NGU method still has the best performance in the Gravitar environment, but our method outperforms the NGU method in the other five environments.

The rewards for the six difficult exploration environments in Atari are relatively sparse. Since the auxiliary agent framework is used in the training, it can effectively help the main agent jump out of the current state when encountering exploration difficulties and continue to perform subsequent exploration steps. When falling into a local optimum during the loss optimization process, the agent must jump out of the current suboptimal state and try to reach a better state as much as possible.

*4.3. Auxiliary Agent Quantity Experiment*

In our proposed framework E2S2RND, the second training phase uses an auxiliary agent for random exploration. However, multiple auxiliary agents can be used at the same time, so we explored the effect of the number of auxiliary agents on the performance of the main agent, and the experimental results are shown in Figure 7. It can be seen from the experimental results that the more the auxiliary agents, the better the exploration speed and final performance of the main agent.

However, in the actual experiment, because the auxiliary agent occupies a large amount of resources, and due to the limitation of the Replay-Buffer size, too many auxiliary agents does not improve the final performance. And because there are too many auxiliary agents, the data in the buffer area may not be updated in time and the auxiliary agents may lag behind, which may lead to performance fluctuations in the learning process of the main agent (the orange curve in Figure 7 has down and up swings). Therefore, when choosing the number of auxiliary agents, many aspects need to be considered.

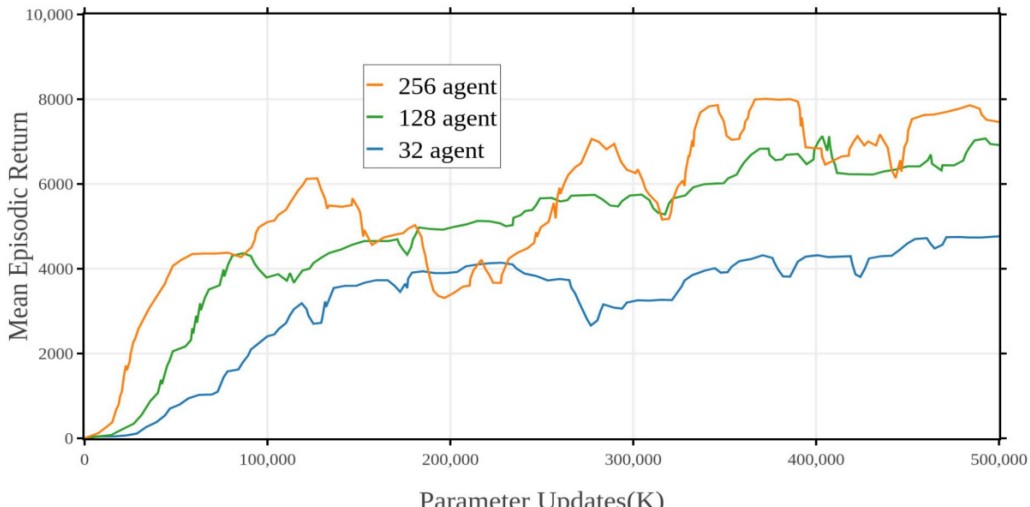

**Figure 7.** Experimental results in Montezuma's Revenge. We used three numbers of auxiliary agents for comparative experiments, including 256 agents (orange), 128 agents (green), and 32 agents (blue), where the horizontal axis represents the model parameter update step, and the vertical axis represents the average return reward of the main agent.

### 4.4. Ablation Experiment

Here we also performed a simple ablation of the optimized feature extraction module and the auxiliary agent training framework in the Montezuma environment, and the experimental results are shown in Table 2. By comparing with the baseline method RND, two improvements in this paper are analyzed.

**Table 2.** Ablation experiments in the Montezuma environment. RND indicates the baseline method, and the other three are ablated on the basis of RND; ESSRND indicates that only the optimized feature extraction module is added; "RND + auxiliary agent" indicates that only the auxiliary agent training framework is used; E2S2RND indicates that the optimized feature extraction module and auxiliary agent training framework were used. Raise ratio is the percentage improvement compared to RND.

|  | Montezuma | Raise Ratio (%) |
|---|---|---|
| RND [29] | 8152 |  |
| ESSRND | 9237 | 13.3 |
| RND + auxiliary agent | 8724 | 7.0 |
| E2S2RND | **10,436** | **28.0** |

In Table 2, using only the optimized feature extraction module, only using the auxiliary agent training framework, or using both combined improves the results relative to the baseline RND by 13.3%, 7.0%, and 28.0%, respectively.

Only using the optimized feature extraction module improves the results by 6.3 percentage points compared with only using the auxiliary agent training framework, in which can be seen that the former contributes more to the method in this paper. Although the auxiliary agent can alleviate the problem of the exploration dilemma, in the face of the ever-changing white noise environment, only using the auxiliary agent is far from enough to solve the impact of the environment changing at any time, and the optimized feature extraction module can learn from the encoder phase to solve this problem.

And the results of using only one improvement point for the experiment are smaller than the results of using both at the same time. It can be concluded that although using one of the parts alone can improve the model effect, the degree of improvement is less than the combination of the two. Therefore, the two improvements we proposed can both improve

the performance of the model in difficult exploration environments, and the effect of using both at the same time is better.

## 5. Conclusions

We present a novel and sophisticated approach, namely E2S2RND, which employs an intricate error prediction mechanism hinged upon finely-tuned features. Additionally, we harness the power of an auxiliary agent to facilitate comprehensive training. By leveraging the predictive errors as a means to engender intrinsic rewards, we introduce a groundbreaking paradigm for refining the exploration methodology plagued by challenges such as separation and derailment. Through meticulous enhancements to the RND method coupled with the incorporation of an optimized feature extraction module, we elevate the resilience of the RND technique against the adversities posed by white noise interferences. Ultimately, the experimental section substantiates the formidable efficacy of our proposed method.

Similarly, since our method is mainly used to solve the problem of exploration difficulty in environments with sparse rewards, the results obtained by our method in other environments with dense rewards are not satisfactory, compared with the current optimal method. There is a big gap.

In future research, we also plan to extend the method of this paper to reward intensive environments, so that it can have a larger scope of application and perform well in different types of environments.

**Author Contributions:** Conceptualization, X.L.; methodology, Q.L.; validation, T.G.; formal analysis, Y.L.; data curation, Q.L.; writing—original draft preparation, T.G.; writing—review and editing, X.L.; visualization, Q.L. All authors have read and agreed to the published version of the manuscript.

**Funding:** This work was supported by the National Science Foundation of China (No. 61972065); Shanghai Science and Technology Plan Project (No. 23692106100, 23DZ2204500).

**Data Availability Statement:** The data presented in this study are available on request from the corresponding author.

**Conflicts of Interest:** The authors declare no conflict of interest.

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
