# Peer review of "Optimized Feature Extraction for Sample Efficient Deep Reinforcement Learning"

_electronics, doi:10.3390/electronics12163508_

Round 1

Reviewer 1 Report

Please replace clear results graphs into this manuscript. In addition, in order to present your contributions, please add compariable experiments with current SOTA.

Please ask native speaker to further improve this manuscript.

Author Response

At present, the best methods are R2D2 method and NGU method in the table, and the E2S2RND method in this paper is mainly compared with the results of the above two methods. In addition, this paper is based on the improvement of RND method, so the results of RND method are also compared with this paper as baseline method.

Thank you for your valuable comments, which are of great help to our work.

Reviewer 2 Report

The Paper has a scientific contribution and represent progress in the research of the subject field.

The introductory part of the Paper should be expanded in terms of reviewing the significance of the research. 

The Results are correctly presented and explained as well as the discussion.

The conclusion is acceptable, although the Paper should be supplemented with more references from theresearch subject. 

Author Response

1、The significance of the research is expanded in the introductory part of the paper, highlighted in orange.

2、Three references have been added in the introduction section, which are about sparse rewards and studies on encouraging exploration by adding noise, highlighted in orange.

Thank you for your valuable comments, which are of great help to our work.

Reviewer 3 Report

see the attached file for the report. 

    Check the writing of the manuscript. There are some typos and grammatical errors which need to be fixed.

Reviewer 4 Report

The authors presented an innovative technique aimed at addressing suboptimal sample efficiency within deep reinforcement learning. The proposed method, E2S2RND, builds upon the RND (Random Network Distillation) approach and introduces a refined feature optimization module known as ESSRND. The authors demonstrate that this feature extraction module significantly enhances the resilience of the RND method against white noise interference, which is a challenging issue in reinforcement learning. One of the key contributions of the E2S2RND method is the introduction of an ancillary agent training paradigm. By leveraging predictive errors to generate intrinsic rewards, this paradigm effectively mitigates the problems of detachment and derailment commonly encountered in exploration methodologies relying on intrinsic rewards. Consequently, the agent's capacity for comprehensive exploration within environments characterized by sparse reward distribution is greatly improved. The authors conducted comparative experiments using challenging exploration problem scenarios, which demonstrated the efficacy of the optimized feature extraction module. Additionally, they conducted a comprehensive performance analysis within the Atari 2600 experimental setting, where the E2S2RND method showed noteworthy advancements in performance, achieving superior outcomes across select experimental environments.

The research work is technically sound and shows improvement in optimizing the feature extraction in deep reinforcement learning. However, there is room for improvement.

  1. Introduction: Provide a short (2-3 sentences) general and broad introduction, and then state the research motivation.
  2. I want to explicitly see a subsection on the need to optimize feature extraction by comparing the disadvantages of traditional methods.
  3. Also, explicitly mention the research questions of the paper. This way, you can grab the reader’s attention.
  4. The authors explained the method and the results of the research. However, they lacked an explanation of the implementation setup and did not provide the details of how they derived the results. For example, provide the details of the dataset used, the programming languages used to implement the algorithms, etc.
  5. Did the authors use any mechanism to validate their results?
  6. What are the limitations of your research?
  7. Provide some future areas of research.

Author Response

1、At the end of the introduction, a short summary and research motivation are added, highlighted in orange.

2、The baseline RND method of this paper is introduced in Section 2.4, and the content of the optimized feature extraction module is introduced in detail in Section 3.1, and the performance of the optimized feature extraction module in the experiment is listed separately in the experimental part of this paper, which can be proved by experiments Necessity to join this module.

3、The research questions of this paper are summarized in the introduction and presented in the second paragraph, highlighted in green.

4、The dataset for this paper is briefly described in Section IV, using six reward-sparse environments and one maze environment in Atari.

5、The experimental verification is mainly obtained by comparing the experimental charts, and the effectiveness of the method in this paper is verified by ablation in the table.

6、In the conclusion section, the limitations of the method in this paper are shown, highlighted in green.

7、In the conclusion section, our future research content is shown, highlighted in green.

Thank you for your valuable comments, which are of great help to our work.

Round 2

Reviewer 1 Report

Please add an ablation experiment. In addition, you may have some references like the below papers.

1. Zhu B, Tao X, Zhao J, et al. An integrated GNSS/UWB/DR/VMM positioning strategy for intelligent vehicles[J]. IEEE Transactions on Vehicular Technology, 2020, 69(10): 10842-10853.

2. Li X, Tao X, Zhu B, et al. Research on a simulation method of the millimeter wave radar virtual test environment for intelligent driving[J]. Sensors, 2020, 20(7): 1929.

3. Tao X, Zhu B, Xuan S, et al. A multi-sensor fusion positioning strategy for intelligent vehicles using global pose graph optimization[J]. IEEE Transactions on Vehicular Technology, 2021, 71(3): 2614-2627.

4. Bing Z H U, Xiao-wen T A O, Jian Z, et al. Two-stage UWB positioning algorithm of intelligent vehicle[J]. Journal of Traffic Engineering, 2021, 21(2): 256-266.

5. Zhao, J., Du, J., Zhu, B., Wang, Z., Chen, Z., & Tao, X. (2022). Intelligent vehicle longitudinal cruise control based on adaptive dynamic sliding mode control. Automotive Engineering, 44(1), 8-16.

6. Zhu, B., Wang, Z., Zhao, J., & Tao, X. (2021). Automatic parking space recognition method based on laser radar bounding boxes. Proceedings of the 2021 Annual Conference of the China Society of Automotive Engineers (Vol. 1).

Please ask native speakers to read the manuscript carefully.

Author Response

1、Added ablation experiments in subsection 4.4, highlighted in purple.

2、References 2, 3 and 5 that you recommend are cited in the introduction section, highlighted in purple.

Thanks again for your valuable comments.

Reviewer 3 Report

If the method in this paper has not been applied to the actual environment. Then there is no point to put effort into these methods. Although you people have answered my all queries.  

Its ok. 

Author Response

Thanks for the comment. All deep reinforcement learning algorithms are firstly tested in benchmark datasets when publication, and then tried out in real world environments by engineers. Our method follows the deep reinforcement learning algorithm reseach convention and is tested in benchmark datasets, which are simulations of real applications. 

Reviewer 4 Report

The authors have improved the manuscript. However, I strongly recommend proofreading one more time.

Author Response

The content of the article was proofread again.